# MTF
## Aplikacja do planowania podróży między wieloma miastami różnymi środkami transportu

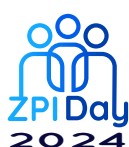

**Autorzy**: Agata Belczyk[®] · Uladzislau Partnou[®] · Zlata Ranchukova[®] · Sara Studzińska[®]
**Opiekun:** Anita Walkowiak-Gall[®]

**Streszczenie**

Aplikacja **MTF - MultiTripFinder** została zaprojektowana, aby uprościć planowanie podróży obejmujących wiele miast, z wykorzystaniem różnych środków transportu: autobusów, pociągów i samolotów. Dzięki integracji funkcjonalności różnych platform podróżniczych, użytkownicy mogą łatwo znaleźć najtańsze lub najszybsze opcje podróży po Europie, uwzględniając własne preferencje dotyczące kosztu i czasu. Aplikacja sugeruje optymalną kolejność odwiedzenia miast na podstawie zdefiniowanych przez użytkownika kryteriów i prezentuje szczegóły dotyczące całej podróży oraz transferów pomiędzy poszczególnymi miastami. Aplikacja zarejestrowanym użytkownikom umożliwia zapisanie wyszukiwań oraz podróży do późniejszego wykorzystania. Dodatkowo, użytkownicy mogą zarządzać zapisanymi podróżami i wyszukiwaniami, a także stosować różnorodne filtry, co pozwala na szybkie znalezienie interesujących ich wpisów. Aplikacja oszczędza czas i pieniądze, oferując wygodne narzędzie do planowania podróży.

# 1 WSTĘP

## 1.1 Opis problemu i jego tło

Planowanie podróży obejmujących wiele miast jest skomplikowanym procesem, zwłaszcza gdy konieczne jest połączenie różnych środków transportu, takich jak autobusy, pociągi i samoloty. Choć istnieje wiele aplikacji umożliwiających wyszukiwanie tras między dwoma punktami, planowanie bardziej złożonej podróży, obejmującej wiele miejsc i różnorodne kryteria, wymaga korzystania z różnych platform. Dodatkowo, znalezienie najtańszej lub najszybszej opcji często wiąże się z ręcznym porównywaniem ofert z różnych źródeł, co jest czasochłonne i nieefektywne.

## 1.2 Cele projektu

Celem projektu było stworzenie aplikacji **MTF - MultiTripFinder**, która uprości proces planowania podróży poprzez:

- Integrację różnych środków transportu w jednym systemie.

- Uwzględnianie preferencji użytkownika, takich jak najniższy koszt podróży, najszybszy czas transferów czy preferowany środek transportu.

- Proponowanie optymalnej kolejności odwiedzania miast na podstawie wybranych kryteriów.

- Zapisywanie otrzymanych tras podróży do późniejszego wglądu na liście podróży.

- Zapisywanie wyszukiwań wraz z określonymi kryteriami, co umożliwia ich ponowne wykorzystanie w przyszłości.

Aplikacja przynosi istotne korzyści zarówno użytkownikom indywidualnym, jak i branży turystycznej. Użytkownicy oszczędzają czas i pieniądze dzięki zautomatyzowanemu planowaniu i optymalizacji tras, a także mają możliwość łatwego zarządzania swoimi wyszukiwaniami i podróżami.

## 1.3 Analiza istniejących rozwiązań

Istniejące rozwiązania, takie jak Google Maps, Google Flights, Skyscanner czy Rome2Rio, oferują wsparcie w planowaniu podróży, jednak mają istotne ograniczenia. Google Maps koncentruje się na wyszukiwaniu tras pomiędzy dwoma punktami, jednak nie oferuje kompleksowego planowania podróży obejmujących wiele miast. Google Flights i Skyscanner skupiają się wyłącznie na połączeniach lotniczych, bez integracji z innymi środkami transportu. Rome2Rio umożliwia planowanie tras między wieloma miastami, łącząc różne rodzaje transportu, ale nie wspiera optymalizacji tras wieloetapowych na podstawie kryteriów takich jak koszt, czas czy preferencje użytkownika dotyczące czasu spędzonego w każdym z miejsc w podróży.

Nasz projekt wyróżnia się możliwością optymalizacji wieloetapowych tras z uwzględnieniem preferencji użytkownika. W przeciwieństwie do Rome2Rio, MTF nie tylko przedstawia dostępne opcje transportu, ale także sugeruje optymalną kolejność odwiedzenia miast. Dodatkowo aplikacja umożliwia zapisywanie podróży i wyszukiwań, a także późniejsze ich przeglądanie, edytowanie i zarządzanie.

## 1.4 Założenia projektowe i wyzwania

Założeniem projektu było stworzenie aplikacji, która łączy funkcjonalności różnych platform podróżniczych oraz umożliwia optymalizację tras według wybranych kryteriów, takich jak czas podróży czy koszt. W ramach realizacji projektu zdecydowaliśmy się na wykorzystanie Spring Boot do stworzenia backendu aplikacji oraz React do budowy interfejsu użytkownika. Dzięki integracji z zewnętrznymi API, takimi jak Amadeus [1] i OpenTripFinder [2], aplikacja umożliwia zaawansowane wyszukiwanie i planowanie podróży. API Amadeus wykorzystano do wyszukiwania połączeń lotniczych, a OTP do wyznaczania tras transportu naziemnego, obejmujących połączenia kolejowe i autobusowe.

Jednym z wyzwań projektu było pozyskiwanie danych w formacie General Transit Feed Specification dla OTP. GTFS zawiera informacje o trasach transportu publicznego, takich jak ścieżki między połączeniami czy koszty podróży. Nie wszystkie kraje dostarczają te dane, co ograniczyło dostępność funkcjonalności do wybranych regionów. W efekcie musieliśmy skoncentrować się na Europie, gdzie dostępność GTFS jest większa. Mimo to, nawet w Europie, brak danych GTFS dla niektórych połączeń naziemnych ogranicza kompletność wyników.

Największym wyzwaniem było opracowanie algorytmu zdolnego do przetwarzania skomplikowanych zapytań i analizowania wyników w sposób szybki oraz efektywny. Algorytm ten musiał być w stanie obsługiwać dane z różnych zewnętrznych źródeł i optymalizować trasy zgodnie z preferencjami użytkownika. Podczas prac nad rozwiązaniem analizowaliśmy różne podejścia i struktury, aby wybrać te, które najlepiej odpowiadają złożoności algorytmu oraz zapewniają najlepsze rezultaty.

## 2 WYNIKI PRACY

Projekt aplikacji MTF zaowocował stworzeniem narzędzia, które spełnia wymagania i założenia biznesowe oraz posiada szeroki zakres funkcjonalności. W podstawowej wersji aplikacja umożliwia niezalogowanym użytkownikom korzystanie z kluczowej funkcji, jaką jest wyszukiwanie podróży. Bardziej zaawansowane funkcje, takie jak zarządzanie listą wyszukiwań i podróży, są dostępne wyłącznie dla zalogowanych użytkowników. Dodatkowo, administratorzy mają dostęp do specjalnego widoku zawierającego pełną listę użytkowników wraz z podziałem na role oraz możliwością zarządzania użytkownikami.

### 2.1 Logowanie za pomocą OAuth2 z Google

Aby uprościć proces logowania i rejestracji, umożliwiliśmy logowanie za pomocą OAuth2 Google [3]. Dzięki temu użytkownicy mogą zalogować się szybko i wygodnie, korzystając ze swojego konta Google. Rozwiązanie to nie tylko zwiększa wiarygodność aplikacji, ale także eliminuje konieczność ręcznego wprowadzania danych podczas rejestracji. Po pierwszym logowaniu aplikacja automatycznie tworzy konto użytkownika w bazie danych, przypisując mu domyślną rolę użytkownika.

To podejście zapewnia wysoki poziom bezpieczeństwa i oddelegowuje odpowiedzialność za autentykację do Google, co upraszcza proces logowania dla użytkowników w porównaniu z tradycyjnymi metodami opartymi na hasłach.

## 2.2 Wyszukiwanie podróży

Funkcjonalność wyszukiwania podróży jest kluczowym elementem aplikacji, umożliwiającym użytkownikom zaplanowanie kompleksowych tras między wieloma miastami.

Aplikacja pozwala użytkownikowi na szczegółowe określenie kryteriów wyszukiwania podróży, takich jak:

- **Miejsce początkowe i miejsce końcowe** – miejsca, z których użytkownik zaczyna i kończy swoją podróż.

- **Miejsca do odwiedzenia** - miasta, które użytkownik chce odwiedzić, wraz z możliwością określenia preferowanej liczby dni i godzin pobytu w każdym z nich.

- **Liczba pasażerów** - użytkownik może określić liczbę pasażerów w podróży. Informacja ta przyda się do oszacowania kosztów podróży.

- **Datę rozpoczęcia podróży** - data, w której użytkownik chce, aby jego podróż się rozpoczęła.

- **Maksymalną liczbę dni na całą podróż** - maksymalna liczba dni jaką użytkownik może spędzić w podróży, uwzględnia czas spędzony w każdym z miejsc do odwiedzenia oraz czas poświęcony na transfery z jednego miejsca do drugiego.

- **Preferowany środek transportu** - użytkownik może wybrać preferowany środek transportu spośród trzech podanych: autobus, pociąg oraz samolot.

- **Kryterium optymalizacji** - użytkownik ma możliwość wyboru kryterium, na podstawie którego zostanie wybrana najlepsza trasa. Może zdecydować się na kryterium czasu, które priorytetowo traktuje najkrótszy czas transferów, lub kryterium kosztu, które skupia się na znalezieniu najtańszych tras pod względem kosztów transferów.

Do wyboru miejsca początkowego, miejsc do odwiedzenia oraz miejsca końcowego, aplikacja wykorzystuje funkcjonalność autouzupełniania oferowaną przez Google Places API [4]. Umożliwia to użytkownikowi szybkie i precyzyjne wybieranie lokalizacji, a jednocześnie zapewnia spójność danych, które są przetwarzane przez system.

W przypadku wpisania niepoprawnych danych np. brak miejsca startowego czy data z przeszłości jako data rozpoczęcia podróży, aplikacja wyświetla odpowiedni komunikat o błędzie.

Po wprowadzeniu danych użytkownik uruchamia proces wyszukiwania za pomocą przycisku. W rezultacie otrzymuje listę kilku proponowanych tras podróży. Każda podróż zawiera szczegółowe informacje o odwiedzanych miejscach i transferach pomiędzy nimi, w tym:

- **Adres, data i godzina wyjazdu z danego miejsca**,

- **Koszt transferu**,

- **Czas trwania transferu**,

- **Adres, data i godzina przyjazdu do danego miejsca**,

- **Środek transportu**,

- **Nazwa przewoźnika**

Dodatkowo aplikacja prezentuje ogólne informacje o całej podróży, takie jak:

- **Całkowity czas trwania podróży**,

- **Czas trwania wszystkich transferów**,

- **Całkowity koszt podróży**.

## 2.3   Algorytm wyszukiwania podróży

Naszym celem było stworzenie algorytmu, który znajdzie optymalną trasę między miastami, uwzględniając nie tylko odległości, ale także czas pobytu w każdym z miast oraz dostępność różnych środków transportu.

W procesie wyszukiwania, algorytm korzysta z zewnętrznych źródeł danych, takich jak:

· **Amadeus API** [1] – w celu pozyskania informacji o trasach lotniczych, takich jak godziny lotów, dostępność połączeń, czas transferów oraz koszty biletów.

· **OpenTripPlanner** [2] – dla tras naziemnych, takich jak połączenia autobusowe i kolejowe, aplikacja wykorzystuje OTP, który dostarcza danych o trasach, przystankach, czasie przejazdu i kosztach.

· **Nominatim API** [5] - w celu przekształcania nazw miejsc na ich współrzędne geograficzne, które następnie mogą zostać użyte jako dane wejściowe do algorytmu.

Po wysłaniu odpowiednio zdefiniowanych zapytań do każdego z tych źródeł, serwis obsługujący algorytm otrzymuje dane dotyczące wszystkich możliwych połączeń pomiędzy miastami. Na podstawie otrzymanych danych tworzy graf, w którym:

· Wierzchołki to miasta.

· Krawędzie w grafie reprezentują połączenia między miastami, które posiadają różne parametry, takie jak czas wyjazdu, czas przyjazdu, cena, czas transferu oraz typ i nazwa środka transportu (np. pociąg PKP Intercity lub samolot WizzAir).

Do wyszukiwania optymalnych tras w grafie wybraliśmy algorytm przeszukiwania w głąb (DFS - Depth-First Search), który zapewnia najbardziej dokładne rozwiązanie, mimo że jego złożoność obliczeniowa jest stosunkowo wysoka. Wprowadziliśmy jednak szereg ograniczeń, które pozwalają na efektywniejsze działanie algorytmu.

· Rozważamy tylko krawędzie, które pasują do określonych ram czasowych, co pozwala na znaczną redukcję liczby przetwarzanych połączeń.

· W trakcie przeszukiwania, jeśli koszt analizowanej (części) ścieżki przekracza koszt już znalezionych najlepszych (całych) tras, przerywamy dalsze badanie tej gałęzi, co eliminuje nieoptymalne rozwiązania i pozwala na szybsze działanie algorytmu.

· Rozważamy maksymalnie 6 miast, co tworzy graf o 6 wierzchołkach, który można efektywnie analizować za pomocą algorytmów takich jak DFS. Aby uniknąć nadmiernej złożoności grafu, do każdej pary miast przypisujemy optymalną liczbę krawędzi, co zwiększa szanse na znalezienie tras spełniających wszystkie kryteria, jednocześnie ograniczając powielanie zbliżonych połączeń.

## 2.4   Zarządzanie listą wyszukiwań oraz listą podróży

Po otrzymaniu wygenerowanej trasy podróży, zalogowani użytkownicy mogą zapisywać wyszukiwania, co umożliwia ich ponowne wykorzystanie w przyszłości bez konieczności wprowadzania tych samych danych. Użytkownik ma także możliwość zapisywania otrzymanych tras podróży. Zarządzanie tymi listami jest kluczowym elementem, który ułatwia organizację podróży i umożliwia szybki dostęp do zapisanych danych.

### Lista wyszukiwań

Na liście zapisanych wyszukiwań wyświetlane są wszystkie istotne parametry, takie jak zastosowane kryteria wyszukiwania, miejsca oraz data zapisania. Najnowsze wyszukiwania pojawiają się na początku listy, co ułatwia szybki dostęp do nich.

Obok listy znajduje się funkcjonalny filtr, umożliwiający wygodne przeszukiwanie zapisanych wyszukiwań. Użytkownik może filtrować wyniki według środka transportu, kryterium optymalizacyjnego, tagów oraz zakresu dat, w których dane wyszukiwanie zostało zapisane. Dzięki temu użytkownik może szybko odnaleźć konkretne zapytania, dostosowane do jego potrzeb.

Dodatkowo, użytkownik ma możliwość edytowania nazwy zapisanego wyszukiwania oraz przypisania do niego tagów. Istnieje również opcja usunięcia zapisanego wyszukiwania, a także możliwość łatwego przejścia do strony wyszukiwania podróży, gdzie dane zapytanie jest automatycznie uzupełnione, co umożliwia szybkie przeprowadzenie podobnego wyszukiwania.

**Lista podróży**

Użytkownik może zapisać pełną trasę podróży z unikalną nazwą do listy podróży. Na liście podróży znajdują się wszystkie zapisane trasy, posortowane według daty zapisania. Po kliknięciu na daną podróż, użytkownik ma dostęp do szczegółów, w tym informacji o poszczególnych transferach i miejscach. Lista podróży może być filtrowana według daty oraz tagów, co ułatwia szybkie znalezienie konkretnych tras. Podobnie jak przy wyszukiwaniach, użytkownik ma możliwość edytowania nazw zapisanych podróży oraz usuwania podróży z listy.

**Tagi**

MTF umożliwia dodawanie, edytowanie i usuwanie tagów zarówno do podróży, jak i wyszukiwań, co ułatwia organizację danych. Użytkownik ma pełną kontrolę nad tagami, może je zmieniać bez modyfikowania całych wpisów oraz przypisywać wiele tagów do jednej podróży lub wyszukiwania. Istnieje również możliwość usunięcia tagu z pojedynczego wpisu, a także globalnego usunięcia go ze wszystkich podróży lub wyszukiwań, w których był używany.

## 2.5 Funckjonalności administratora

Administrator w aplikacji ma dostęp do panelu zarządzania, który pozwala na kontrolę nad kontami użytkowników. Główne funkcjonalności dostępne dla administratora to:

- **Przeglądanie listy**: Administrator ma możliwość przeglądania pełnej listy użytkowników, w tym ich e-maili oraz przypisanych ról. Dzięki temu może monitorować użytkowników, którzy korzystają z aplikacji.

- **Usuwanie użytkowników**: W razie potrzeby, administrator ma uprawnienia do usuwania kont użytkowników. Może usunąć zwykłych użytkowników, co pozwala na utrzymanie porządku i bezpieczeństwa w systemie.

## 2.6 Metryki

W ramach oceny skuteczności i użyteczności aplikacji zaplanowaliśmy zebranie opinii użytkowników, aby lepiej zrozumieć ich doświadczenia i preferencje. Planujemy przeprowadzenie badań z użytkownikami, takich jak ankiety, wywiady czy oceny opisowe, które pozwolą na weryfikację aplikacji pod kątem jej funkcjonalności, wydajności oraz intuicyjności interfejsu.

# 3 WYKORZYSTANE TECHNOLOGIE

Do stworzenia backendu aplikacji zastosowaliśmy framework **Spring Boot** [6], który umożliwił nam efektywne zarządzanie konfiguracjami oraz implementację i udostępnienie API zgodnego ze specyfikacją **OpenAPI** [7]. Wykorzystaliśmy **OAuth2** [3], aby zapewnić bezpieczne uwierzytelnianie użytkowników z użyciem zewnętrznych dostawców tożsamości (Google) oraz JWT do umożliwienia zarządzanie sesjami użytkowników. Do implementacji zabezpieczeń, w tym autoryzacji i autentykacji, użyliśmy **Spring Security**.

Nowoczesny i interaktywny interfejs użytkownika zaimplementowaliśmy wykorzystując framework **React** [8], bibliotekę **Mantine** [9] oraz język **TypeScript**.

Algorytm został zaimplementowany jako niezależny serwis w **Pythonie** [10] z wykorzystaniem frameworka **FastAPI** [11]. Python, dzięki swojej wydajności w przetwarzaniu danych i bibliotece **NetworkX** [12] do pracy z grafami, doskonale sprawdza się w tego typu zadaniach. FastAPI, będący jednym z najszybszych frameworków w Pythonie, zapewnia wysoką wydajność, co jest kluczowe przy obsłudze czasochłonnych zapytań do zewnętrznych źródeł oraz przy wykonywaniu przeszukiwania grafu metodą DFS.

Do przechowywania danych aplikacji wykorzystaliśmy **MySQL** [13], który oferuje stabilne i wydajne zarządzanie bazą danych. Aplikację wdrożyliśmy i utrzymujemy na platformie **AWS** [14] z wykorzystaniem konteneryzacji za pomocą **Docker**. Każda z tych technologii została wybrana, aby zapewnić bezpieczeństwo, elastyczność i łatwość w utrzymaniu aplikacji.

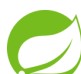 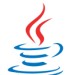 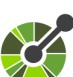 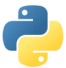 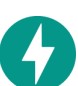 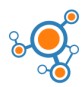 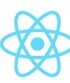 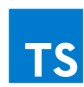 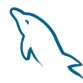 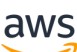 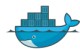

## 4  WNIOSKI

Projekt MTF upraszcza proces planowania podróży, oferując użytkownikom indywidualnym oraz branży turystycznej narzędzie, które integruje różne środki transportu w jednym systemie. Kluczowym osiągnięciem projektu jest stworzenie intuicyjnej aplikacji, która wyszukuje dostępne połączenia i układa je w odpowiedniej kolejności, tworząc optymalną trasę podróży dla użytkownika, spełniająca zdefniowane przez niego kryteria. Możliwość zapisywania wyszukiwań oraz tras podróży umożliwia zalogowanym użytkownikom wygodne zarządzanie planami podróży, co pozwala oszczędzać czas i pieniądze.

Największym sukcesem projektu jest wdrożenie funkcji, które łączą personalizację, wygodę użytkowania i optymalizację tras, wyróżniając aplikację na tle istniejących rozwiązań.

W przyszłości możliwy jest rozwój aplikacji o dodatkowe funkcjonalności, takie jak dodanie większej liczby miejsc startowych i końcowych, wprowadzenie zniżek dla pasażerów, a także opcja dzielenia się zapisanymi podróżami i komentowania ich. W przypadku brakujących danych o kosztach lub trasach, aplikacja mogłaby automatycznie pobierać informacje z różnych zewnętrznych API (np. API dostawców transportu publicznego, API lotów), aby wypełnić luki.

W przyszłej wersji aplikacji możliwa jest także dalsza optymalizacja algorytmu, która obejmie m.in. bardziej zaawansowane techniki odcinania nieoptymalnych tras, dynamiczne dopasowywanie metod transportu w zależności od warunków (np. odległość między miastami) oraz dodatkowe mechanizmy zapobiegające nadmiernemu obciążeniu zewnętrznych API, co pozwoli na szybsze i bardziej efektywne wyszukiwanie tras podróży.

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
