# OpenReview forum: "MultiTripFinder - Aplikacja do planowania podróży między wieloma miastami różnymi środkami transportu"
_pwr.edu.pl/Wrocław_University_of_Science_and_Technology/2024/ZPI_Day — Wrocław University of Science and Technology 2024 ZPI Day Submission_

### Official Review · Reviewer_ukLD · 2024-12-04
**Nowoczesna i przyszłościowa koncepcja**

**Confidence:** 4
**Significance Of Results:** 5
**Overall Quality:** 5

**Compliance With Template:**

5: Very High Quality – The article contains all the required sections, which are written in a very detailed, clear, and error-free manner. The structure is professional and meets expectations, and the content adheres to the highest substantive and formal standards.

**Description Of Results:**

5: Very High Quality – The results are described in detail, clearly and comprehensively, supported by thorough evaluation, analysis, and convincing usage examples. The description meets the highest substantive standards.

**Feedback On Consistency:**

Wskazany projekt stanowi ciekawe rozwiązanie w planowaniu tras podróży z wykorzystaniem niemal wszystkich możliwych środków transportu publicznego na dużych odległościach. Analiza problemu jest spójna i logiczna. Wskazane cele zostały jasno określone. Projekt opisany jest prawidłowo zaczynając od celu przez analizę dostępnych rozwiązań oraz możliwości pozyskiwania potrzebnych danych. Opis zastosowanego rozwiązania  jest przejrzysty i zrozumiały.

**Potential For Development:**

Projekt wykazuje znaczny potencjał rozwojowy. Przede wszystkim można go rozbudować o informacje związane z infrastrukturą docelowych miejsc podróży (hotele, restauracje znajdujące się w bezpośrednim otoczeniu stacji lub lotniska), oraz o informacje dotyczące opóźnień lub transportu zastępczego.

**Project Nature Evaluation:**

Projekt obejmuje wykorzystanie narzędzi informatycznych do stworzenia nowego oprogramowania. Oprogramowanie bazuje na współpracy z danymi dostępnymi z zewnętrznych źródeł. Wykorzystuje metodę matematyczną do wyboru najkorzystniejszego rozwiązania prezentowanego odbiorcy. Projekt wykazuje wysoki stopień użyteczności.

**Technical Language Precision:**

5: Very High Quality – The language is entirely appropriate for a technical report. All terms are used correctly and precisely, and the style is professional, clear, and coherent, without any errors or ambiguities.

---

### Official Review · Reviewer_RWw7 · 2024-12-07
**Recenzja MultiTripFinder**

**Confidence:** 4
**Significance Of Results:** 4
**Overall Quality:** 4

**Compliance With Template:**

4: High Quality – The article contains all the required sections, which are well-written and substantively correct, although minor errors or shortcomings may be present. The overall structure is clear and coherent.

**Description Of Results:**

5: Very High Quality – The results are described in detail, clearly and comprehensively, supported by thorough evaluation, analysis, and convincing usage examples. The description meets the highest substantive standards.

**Feedback On Consistency:**

Analiza problemu przedstawiona w pierwszej części artykułu, jest precyzyjnie i w pełnym zakresie rozwinięta w części opisującej rezultaty projektu. Podsumowanie nawiązuje w pełni do problemów poruszonych we Wstępie i Wynikach pracy.

**Potential For Development:**

Projekt wydaje się być innowacją na rynku, ma duży potencjał wdrożenia komercyjnego. Autorzy przede wszystkim chcą pracować nad usprawnieniem algorytmu wyszukiwania połączeń. Zabrakło wskazówek o nowych funkcjonalnościach systemu.

**Project Nature Evaluation:**

Autorzy poprawnie uzasadniają wybór technologii, ale opis projektu/architektury systemu nie jest wystarczający. Brakuje także opisu procesu wytwórczego.

**Technical Language Precision:**

4: High Quality – The language is appropriate for a technical report. Terminology is used correctly, and statements are precise, with only minor shortcomings that do not affect the overall clarity.

---

### Official Review · Reviewer_ta9o · 2024-12-09
**A practical application with some minor problems**

**Confidence:** 4
**Significance Of Results:** 3
**Overall Quality:** 4

**Compliance With Template:**

4: High Quality – The article contains all the required sections, which are well-written and substantively correct, although minor errors or shortcomings may be present. The overall structure is clear and coherent.

**Description Of Results:**

4: High Quality – The results are described in detail and supported by usage examples or evaluations. The description is reliable but may lack full depth of analysis.

**Feedback On Consistency:**

The presentation of the project is consistent, but there are some very important elements missing. As the application deals with optimization problem in real-world situation, as an application for end-users, some serious testing would be required (border cases for the DFS and user testing, preferentially). Also, the DFS is limited to 6 cities, but it is not one of clearly stated business limitations/requirements

**Potential For Development:**

These are clearly stated and OK.

**Project Nature Evaluation:**

It seems a good application of technical skills, but it is missing key element of real world deployment of engineering work - testing.

**Technical Language Precision:**

4: High Quality – The language is appropriate for a technical report. Terminology is used correctly, and statements are precise, with only minor shortcomings that do not affect the overall clarity.

---

### Decision · Program_Chairs · 2024-12-10

Accept (Poster)